# Evolution of Pretreatment Assessment and Direct Acting Antiviral Regimens in Accordance with Upgrading Guidelines: A Retrospective Study in HIV/HCV Coinfected Patients [note 1]

**DOI:** 10.3390/medsci6040081

**Published:** 2018-09-20

**Authors:** Zachary Henry, Jose Armando Gonzales Zamora

**Affiliations:** Division of Infectious Diseases, Department of Medicine, University of Miami, Miller School of Medicine, Miami, FL 33136, USA; zachary.henry@jhsmiami.org

**Keywords:** HIV, chronic hepatitis C, direct acting antivirals, Miami

## Abstract

Since the advent of new direct acting antivirals (DAA), substantial changes in hepatitis C (HCV) treatment guidelines have occurred. However, little is known about how these recommendations have been adopted into clinical practice. We conducted a retrospective review of human immunodeficiency virus (HIV)/HCV coinfected patients treated with DAAs at the Ryan White Clinic of the Jackson Health System in Miami, FL, USA. Our aim was to determine changes in HCV evaluation and treatment patterns in the use of DAAs over a four-year period from January 2014 to December 2017. Data were divided into two periods: period 1 (2014–2015) and period 2 (2016–2017). In comparison with the rest of the cohort, patients in period 2 had a lower frequency of advanced liver disease (24.4% vs. 48.6%, *p* = 0.026) and underwent more elastography (34.1% vs. 2.7%, *p* < 0.001) and less ultrasound (78.0% vs. 97.3%, *p* = 0.011). They were more often treated with ledipasvir/sofosbuvir (85.4% vs. 56.8%, *p* = 0.005) and less often with simeprevir/sofosbuvir (0% vs. 32.4%, *p* < 0.001). Gastrointestinal side effects were reported less frequently (2.4% vs. 18.9%, *p* = 0.017) in this period. In accordance with the updated guidelines, our study demonstrated a growing preference for non-invasive methods to assess fibrosis in recent years. Regarding treatment, there was a clear preference for second generation DAAs in 2016–2017, along with initiation of treatment in the early stages of liver disease.

## 1. Background

Treatment for hepatitis C (HCV) has come a long way since the days when those afflicted by the virus could rely on a mere 50% chance of a cure with a regimen that seemed more like a year of suffering than a cure for a disease [1,2]. For people coinfected with human immunodeficiency virus (HIV), the outlook was especially grim considering the significant drug–drug interactions of previous HCV medications with antiretroviral therapy (ART) and accelerated progression to cirrhosis. Even when compensated, the decision to start treatment in cirrhotic patients demanded a delicate balance between further deferring treatment and the risk of serious side effects with the interferon-based regimens [3]. Since 2014, with the advent of the second-wave first-generation protease inhibitors, and especially the second-generation direct acting antivirals (DAAs), HCV treatment has progressed to include interferon-free regimens that optimize treatment time with safety, efficiency, and tolerability for all 200 million chronically infected, and the 4 million newly infected yearly [3]. As treatment guidelines have evolved to implement these new regimens in accordance with top-tier recommendations, we would hope to see quelled the virus’ ability to propagate itself as the principal agent of hepatocellular carcinoma, cirrhosis, and the need for liver transplant. Several authors have noted these aforementioned improvements in HCV treatment and the DAAs’ high rates of sustained virologic response (SVR). However, they have also discerned that along with ever-evolving treatments and changing guidelines, comes inevitable ambiguity as to the degree to which these new regimens can feasibly be adopted into clinical practice [4]. We acknowledge this matter, but just as crucial is the overall impact on cure rates between the different regimens, especially in those populations underrepresented in clinical trials, such as those with advanced liver disease and/or coinfected with HIV [5].

Also, while the standard of care for assessing a patient’s degree of liver damage has been the liver biopsy, in recent years less invasive measures, with results proven to be comparable in precision to those of liver biopsy, have been introduced. Modalities that utilize serum markers and advanced ultrasound technology, such as the fibrotest and transient elastography, have been conceived for more rapid, and subsequently simpler, means of assessment for liver fibrosis [6,7]. In terms of HCV treatment, many experts recommend that an earlier treatment with DAAs should always be offered whenever possible [8]. This makes sense given not only their superiority to interferon-based regimens, but also in regards to the ease of pretreatment evaluations available now that dictate their usage. Given the above preferences in regards to DAAs, we contribute our own data and findings which are in the same vein as, and complimentary to, the aforementioned literature. Our aim was to determine the changes in HCV evaluation and treatment patterns over a 4-year period using DAAs in HIV/HCV coinfected patients from January 2014 to December 2017.

## 2. Methods

We conducted a retrospective review of HIV/HCV coinfected patients treated with DAAs at the Ryan White Clinic of the Jackson Health System in Miami, FL, USA. The study was approved by the Institutional Review Board (IRB) of the University of Miami, with IRB code 20170867. Data were divided into two periods; period 1 (2014–2015), and period 2 (2016–2017). We compared demographics and clinical variables by treatment period (Table 1). Data were analyzed using SPSS 22 (IBM, New York, NY, USA). The patients from period 2 were compared to the cohort from period 1 using chi-square test for categorical variables and Student’s *t*-test for continuous variables. All tests were two-tailed, and a *p*-value of <0.05 was considered statistically significant. The odds ratios (OR) were calculated with 95% confidence intervals (CI). Being cured was defined as having a documented undetectable HCV RNA at 12-weeks post-treatment (SVR12). We used “intention to treat” and “treated analysis” to evaluate SVR12. In the intention to treat analysis, we included treatment failures, loss to follow-up, and patients who completed HCV treatment, but who did not have their HCV RNA checked at 12-weeks post-treatment. On the other hand, “treated analysis” only included patients who did not achieve SVR12 after adequate completion of therapy. Treatment failure was defined as detectable HCV RNA during or after DAA therapy. To measure the HCV viral load, we used a real-time polymerase chain reaction (PCR) (Roche HCV TaqMan assay, Hoffmann-La Roche, Basel, Switzerland), which quantifies HCV RNA over a range of 15 to 100,000,000 UI/mL. Results below 15 UI/mL were reported as undetectable. In regards to liver disease severity, advanced liver disease was defined as a biopsy fibrosis stage of ≥F3 or transient elastography score of ≥9.5 KPa.

## 3. Results

There were 78 coinfected patients. The male to female ratio was 2:1, with a mean age of 55.6 years (SD ± 7.88 years) and most patients were African American (57.7%). There were 37 (47.4%) patients in period 1 and 41 (52.6%) patients in period 2. Antiretroviral therapy was received by 96.2% of patients. The various regimens used had a base of either tenofovir disoproxil (TDF), or tenofovir alafenamide (TAF) and emtricitabine (FTC), with either a protease inhibitor (PI), non-nucleoside reverse transcriptase inhibitor (NNRTI), or integrase strand transfer inhibitor (InSTI). Other regimens included a base of abacavir (ABC) and lamivudine (3TC), with a PI or InSTI. In period 1, ART regimens of TDF/FTC with either a PI, NNRTI, or InSTI were collectively used 70.2% of the time. In period 2, the most frequently used regimen was TDF/FTC with a PI (29.3%). The mean CD4+ T lymphocyte count was 637.68 cells/uL (SD ± 334.35). Liver biopsies were done on 42.3% of the patients. Of the total cohort, 35.9% had advanced liver disease and 15.4% were cirrhotic. Prior treatment with an interferon regimen was noted in 28.2% of the total cohort. Genotype 1a was the most frequent HCV genotype (60.3%). The mean baseline HCV log10 IU/mL was 6.18 (SD ± 0.76). The DAA regimens used, which included first- and second-generation drugs, were simeprevir/sofosbuvir and paritaprevir/ritonavir/ombitasvir/dasabuvir (PROD), with ribavirin. The strictly second-generation regimens used were ledipasvir/sofosbuvir, elbasvir/grazoprevir and sofosbuvir/velpatasvir. The designated DAA regimens were completed by 73 patients (93.6% of the total cohort), with 5 patients in total (6.4%) being lost to follow-up. Overall, among the cohort, 64 patients achieved SVR12. The cure rate for the intention to treat analysis was 82.05%, while in the treated analysis this increased to 95.5%. Overall, the most frequently reported side effect was fatigue, as reported by ten patients, or 12.8% of the total cohort. Ledipasvir/sofosbuvir was used for the majority of the time in both periods. In comparison with the rest of the cohort, patients in period 2 had a lower frequency of advanced liver disease (24.4% vs. 48.6%, *p* = 0.026), and underwent more elastography (34.1% vs. 2.7%, *p* < 0.001) and less ultrasound (78.0% vs. 97.3%, *p* = 0.011). They were more often treated with ledipasvir/sofosbuvir (85.4% vs. 56.8%, *p* = 0.005), and less often with simeprevir/sofosbuvir (0% vs. 32.4%, *p* < 0.001). Gastrointestinal side effects were reported less frequently (2.4% vs. 18.9%, *p* = 0.017) in this period. In terms of ART, there was a trend towards more frequent use of TAF/FTC with InSTI (9.8% vs. 0%, *p* = 0.05) in period 2, and we noted a decreasing trend in treatment failures (0% vs. 8.1%, *p* = 0.06) (Table 1). 

## 4. Discussion

We believe our study is the first of its kind to evaluate how treatment recommendations in current guidelines have been adopted in clinical practice. We noticed an increasing proportion of patients on ledipasvir/sofosbuvir in period 2 in comparison to period 1 (85.4% vs. 56.8%, *p* = 0.005). The preference for ledipasvir/sofosbuvir in our clinic in 2016–2017 is in accordance with the recommendations of the updated AASLD/IDSA (American Association for the Study of Liver Diseases/Infectious Diseases Society of America) guidelines, in which ledipasvir/sofosbuvir, along with other strictly second-generation DAA regimens, constitute the first line drugs for the treatment of chronic hepatitis C [9]. Another important finding was the predominant use of simeprevir/sofosbuvir in period 1 (32.4% vs. 0%, *p* < 0.001). Of note, during period 1 (2014–2015) the treatment of hepatitis C in HIV-infected individuals was limited to regimens that included first generation DAAs, such as simeprevir/sofosbuvir and PROD with ribavirin. With the approval of ledipasvir/sofosbuvir for the treatment of HIV/HCV coinfected patients in October 2015, the first generation DAA regimens were left aside, and became second line or alternative drugs in the treatment of hepatitis C. By evaluating the treatment patterns in our clinical practice over four years, we observed that the guideline recommendations were being implemented by our HIV/HCV providers in a proper manner. Our study also showed a decreasing trend for g treatment failures in period 2 (0% vs. 8.1%, *p* = 0.006). The reason for this finding was probably the higher effectiveness of ledipasvir/sofosbuvir; however, the design of our study did not allow for an adequate evaluation of this parameter. 

Another interesting trend in period 2 was the increased proportion of patients on TAF/FTC and InSTI, with a *p*-value of 0.05. In November 2016, with the introduction of TAF, the threats of renal injury and bone mineral loss that burdened those on the TDF counterpart were largely alleviated, making TAF-consisting regimens the preferred options for treatment of HIV in clinical practice [10]. Despite the vast breadth of improvement in these therapies in and of themselves, there persists a never-ending concern with drug–drug interactions and adverse effects associated with keeping the ART and DAA regimens running in parallel. For the physician, this presents the problem of having to continuously evaluate a seemingly unavoidable series of “checks and balances” in managing the two regimens, as both must work together in order to keep these already difficult-to-manage patients alive. These bothersome interactions and effects that can present limitations in optimizing care have failed to be completely overcome even today, as another hurdle is presented by the increased serum concentrations of TDF in the presence of ledipasvir. This issue further endangers patients’ kidneys and bone mineral density, but is absent with the substitution of TAF [11]. We were able to successfully begin incorporating the safer alternative of TAF into our ART regimens in period 2, which ensures better compatibility with the newer DAA regimens, as it is imperative that they act in harmony to promote the health of this coinfected population. To say the increase in magnitude for the ART parameter in period 2 is statistically relevant, and not just coincidence, would be an assumption. However, the probability that said assumption is false is low enough that by revamping the study with adequate power from the beginning, we may have been able to show with statistical significance that the use of TAF/FTC with InSTI increased in period 2 and that treatment failures decreased.

One parameter that improved statistically significantly from period 1 to period 2, was the lack of gastrointestinal (GI) side effects noted in the later years with decreased use of simeprevir/sofosbuvir. Aside from being already proven to lack more serious side effects and drug–drug interactions that might impede or even contraindicate its use from the start, the decrease in unpleasant side effects with greater use of strictly second-generation containing regimens likely contributed to increased compliance and a better success rate [12]. 

In terms of pretreatment evaluation of liver fibrosis, we saw more patients undergoing transient elastography in period 2 (2016–2017). Of note, liver biopsy is still considered the gold standard for assessing fibrosis; however, it is less frequently used because of its invasive nature and the risk of complications. In recent years, various noninvasive markers have become more widely available (e.g., Fib-4, fibrosure, transient elastography) and the role of liver biopsy has been limited to cases of uncertain diagnosis or for evaluation of concomitant pathologies [13].

Another interesting finding of our study was the decrease in the number of patients with advanced liver disease in period 2 (*p* = 0.026). The clinical relevance of this parameter is clearly prognostic of the notion that with the new and improved treatment regimens and pretreatment screenings, physicians are more willing and able to bring these chronically-infected patients into care. On the other side, patients are more likely to seek out and accept their own instatement into care earlier. This demonstrates that HCV treatment can be promptly offered to, and welcomed by, patients as a likely cure for a fatal disease. Treatment is no longer pushed upon them as an adjunct to the patients’ and physicians’ mutual acquiescence in reaching the “end of the line” with their chronic disease [14]. 

The ascendance of strictly DAAs over interferon-based regimens has already been well demonstrated and accepted, as they have almost universally been adopted into practice. However, with the accelerated production of the new second generation DAAs, the potential for better patient outcomes with these ever-advancing regimens should quickly downgrade older first-generation containing regimens to the “bottom of the barrel”, so to speak, of the infectious disease specialist’s arsenal of antiviral therapies. We would hope to see more prevalent use of DAAs such as sofosbuvir, ledipasvir, velpatasvir, daclatasvir, and now glecaprevir and pibrentasvir, in all clinical practices. These drugs would allow a higher safety profile, increase ease of administration and decrease treatment duration. On the downside, while their prestige has garnered them significant attention within the medical community, so have their exorbitant costs. Regardless, studies have suggested that these regimens are still cost-effective overall given their overwhelming superiority in clinical efficiency, despite the face value of their monetary requirements, [15,16]. All these factors, along with advancing and less invasive modalities used in the pretreatment evaluations needed to coordinate the regimen most appropriate for a given patient, should promote facilitation of earlier access to and acceptance into care. This will lead to a more limited number of patients seeking treatment while in cirrhosis, and offer hope for a cure that was denied by the regimens containing interferon, ribavirin, and/or early first generation DAAs for those who already have advanced liver disease. Our study was unable to show a statistically significant difference in the rate of cure, rate of therapy completion, rate of loss to follow-up/return for post-treatment, or rate of treatment failure. However, consistent with the aforementioned presumptions, in our Ryan White Clinic from 2016–2017, we saw more patients undergoing elastography for pretreatment evaluations who were more often treated with regimens of second generation DAAs in accordance with the most updated AALD/IDSA guidelines [9], and fewer patients being initiated into care before progressing to advanced liver disease. 

## Figures and Tables

**Table 1 medsci-06-00081-t001:** Clinical and treatment characteristics of human immunodeficiency virus (HIV)/hepatitis C (HCV) coinfected patients and comparison by treatment period (periods 1 and 2).

Variable	2014–2017	Period 1 (2014–2015)	Period 2 (2016–2017)	*p*-Value	OR (95% CI)
	(*n* = 78)	(*n* = 37)	(*n* = 41)		
**Age**	55.64 ± 7.88	56.59 ± 6.46	54.78 ± 8.97	0.31	-
**Sex (male)**	53 (67.9%)	25 (67.6%)	28 (68.3%)	0.95	1.02 (0.65–1.60)
**Race**					
(a) Black	45 (57.7%)	19 (51.4%)	26 (63.4%)	0.28	1.27 (0.81–1.99)
(b) White	13 (16.7%)	9 (24.3%)	4 (9.8%)	0.09	0.54 (0.23–1.26)
(c) Hispanic	20 (25.6%)	9 (24.3%)	11(26.8%)	0.8	1.06 (0.67–1.70)
**CD4 count**	637.68 ± 334.35	672.08 ± 309.82	606.63 ± 355.97	0.39	-
**Received ART**	75 (96.2%)	35 (94.6%)	40 (97.6%)	0.5	1.60 (0.32–8.04)
**ART regimen**					
(a) TDF/FTC + NNRTI	15 (19.2%)	8 (21.6%)	7 (17.1%)	0.61	0.87 (0.48–1.56)
(b) TDF/FTC + PI	21 (26.9%)	9 (24.3%)	12 (29.3%)	0.62	1.12 (0.72–1.76)
(c) TDF/FTC + InSTI	15 (19.2%)	9 (24.3%)	6 (14.6%)	0.28	0.72 (0.37–1.39)
(d) TAF/FTC + InSTI	4 (5.1%)	0 (0%)	4 (9.8%)	0.05	2.0 (1.59–2.51)
(e) ABC/3TC + InSTI	7 (9.0%)	1 (2.7%)	6 (14.6%)	0.07	1.74 (1.19–2.55)
(f) ABC/3TC + PI	4 (5.1%)	3 (8.1%)	1 (2.4%)	0.26	0.46 (0.08–2.56)
(g) Other regimens	9 (11.5%)	5 (13.5%)	4 (9.8%)	0.6	0.83 (0.39–1.78)
**Prior Tx with IFN**	22 (28.2%)	12 (32.4%)	10 (24.4%)	0.43	0.83 (0.50–1.40)
**Liver biopsy**	33 (42.3%)	18 (48.6%)	15 (36.6%)	0.28	0.79 (0.50–1.23)
**Elastography**	15 (19.2%)	1 (2.7%)	14 (34.1%)	<0.001	2.18 (1.59–2.99)
**US**	68 (87.2%)	36 (97.3%)	32 (78.0%)	0.011	0.52 (0.38–0.72)
**Genotype**					
(a) 1a	47 (60.3%)	24 (64.9%)	23 (56.1%)	0.43	0.84 (0.56–1.28)
(b) 1b	25 (32.1%)	10 (27.0%)	15 (36.6%)	0.37	1.22 (0.80–1.86)
(c) Others	6 (7.7%)	3 (8.1%)	3 (7.3%)	0.9	0.95 (0.41–2.18)
**HCV10log**	6.18 ± 0.76	6.11 ± 0.71	6.25 ± 0.80	0.42	-
**Creatinine**	1.05 ± 0.38	1.02 ± 0.34	1.08 ± 0.41	0.51	-
**Advanced liver disease (F3, F4)**	28 (35.9%)	18 (48.6%)	10 (24.4%)	0.026	0.576 (0.34–0.99)
**Cirrhosis**	12 (15.4%)	7 (18.9%)	5 (12.2%)	0.41	0.76 (0.38–1.55)
**HCV treatment**					
(a) Ledipasvir/Sofosbuvir	56 (71.8%)	21 (56.8%)	35 (85.4%)	0.005	2.292 (1.125–4.670)
(b) Simeprevir/Sofosbuvir	12 (15.4%)	12 (32.4%)	0 (0%)	<0.001	-
(c) PROD/RBV	2 (2.6%)	2 (5.4%)	0 (0%)	0.13	-
(d) Elbasvir/Grazoprevir	2 (2.6%)	0 (0%)	2 (4.9%)	0.17	1.95 (1.57–2.43)
(e) Sofosbuvir/RBV	2 (2.6%)	2 (5.4%)	0 (0%)	0.13	-
(f) PROD	1 (1.3%)	0 (0%)	1 (2.4%)	0.34	1.93 (1.55–2.39)
(g) Sofosbuvir/Velpatasvir	1 (1.3%)	0 (0%)	1 (2.4%)	0.34	1.93 (1.55–2.39)
**Tx duration (12 weeks)**	71 (91.0%)	33 (89.2%)	38 (92.7%)	0.59	1.25 (0.52–3.02)
**SVR12 (ITT)**	64 (82.1%)	31 (83.8%)	33 (80.5%)	0.71	0.90 (0.54–1.51)
**SVR12 (treated analysis)**	64/67 (95.5%)	30/33 (90.9%)	34/34 (100%)	0.07	-
**Completed HCV Tx**	73 (93.6%)	35 (94.6%)	38 (92.7%)	0.73	1.38 (0.22–8.76)
**Lost to follow-up**	5 (6.4%)	2 (5.4%)	3 (7.3%)	0.73	1.1153 (0.55–2.43)
**Failed HCV Tx**	3 (3.9%)	3 (8.1%)	0 (0%)	0.06	-
**No labs to assess SVR12**	6 (7.7%)	2 (5.4%)	4 (9.8%)	0.47	1.30 (0.71–2.39)
**Side Effects**					
(a) Headache	3 (3.8%)	2 (5.4%)	1 (2.4%)	0.5	0.63 (0.12–3.14)
(b) GI	8 (10.3%)	7 (18.9%)	1 (2.4%)	0.017	0.22 (0.04–1.38)
(c) Fatigue	10 (12.8%)	6 (16.2%)	4 (9.8%)	0.39	0.74 (0.33–1.62)

OR = odds ratios; CI = confidence interval; CD4 count = CD4+ T lymphocyte count; ART = antiretroviral therapy; TDF = tenofovir disoproxil fumarate; FTC = emtricitabine; NNRTI = non-nucleoside reverse transcriptase inhibitor; PI = protease inhibitor; InSTI = Integrase inhibitor; TAF = tenofovir alafenamide; ABC = abacavir; 3TC = lamivudine; Tx = treatment; IFN = interferon; US = ultrasound; HCV10log = log base 10 of HCV RNA; F3 = fibrosis stage 3; F4 = fibrosis stage 4; PROD = paritaprevir/ritonavir/ombitasvir/dasabuvir; RBV = ribavirin; ITT = intention to treat; GI = gastrointestinal; SVR12 = sustained virologic response at 12-weeks post-treatment. The *p* values that achieved statistical significance (*p* < 0.05) are in bold.

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
