# Peer review of "Evolution of Pretreatment Assessment and Direct Acting Antiviral Regimens in Accordance with Upgrading Guidelines: A Retrospective Study in HIV/HCV Coinfected Patients†"

_medsci, 2018, doi:10.3390/medsci6040081_

Reviewer 1 Report
This article addressed the retrospective analysis of HIV/HCV coinfected patients treated with first and second-generation DAAs into two periods: period 1 (2014-2015) and period 2 (2016-2017). A clear preference for second generation DAAs in 2016-2017, along with initiation of treatment in early stages of liver disease was shown.
The design of the study raises questions. In order to draw conclusions about the greater effectiveness of drugs of the 2nd generation, it is necessary to group patients not by years, but by drugs. It is necessary to carry out this analysis.
The sensitivity of HCV RNA determination, which test systems were used, was not specified.
p.4, line 116. Abbreviations AST, ALT are introduced, and there are no references in the text.
The results repeat the entire table.
Author Response
Dear Reviewer:
Thanks for your positive comments. I added the modifications you suggested. The changes are typed in red in the manuscript. The answers to your queries are the following:
1. Comment: The design of the study raises questions. In order to draw conclusions about the greater effectiveness of drugs of the 2nd generation, it is necessary to group patients not by years, but by drugs. It is necessary to carry out this analysis.
Answer: We agree with the reviewer on this comment. The design of our study does not allow an adequate evaluation of drug effectiveness. We have modified the first paragraph of the discussion to emphasize that the overall objective of our study was to evaluate the changes of treatment patterns over 4 years of direct acting antiviral use in our clinical practice, but not the drug effectiveness. One of the main conclusions of our study was that the implementation of upgrading treatment guidelines was carried out by our providers in an adequate manner. Now the first paragraph of the discussion states the following (lines 124-141):
“We believe our study is the first of its kind to evaluate how treatment recommendations of current guidelines have been adopted into clinical practice. We noticed an increasing proportion of patients on ledipasvir/sofosbuvir in period 2 in comparison to period 1 (85.4% vs 56.8%, p=0.005). The preference for ledipasvir/sofosbuvir in our clinic in 2016-2017 is in agreement with the recommendations of the updated AASLD/IDSA guideline, in which ledipasvir/sofosbuvir along with other strictly second-generation DAA regimens constitute the first line drugs for the treatment of chronic hepatitis C [9]. Another important finding was the predominant use of simeprevir/sofosbuvir in period 1 (32.4% vs 0%, p<0.001). Of note, during period 1 (2014-2015), the treatment of hepatitis C in HIV-infected individuals was limited to regimens that included first generation DAA, such as simeprevir/sofosbuvir, and PROD with ribavirin. With the approval of Ledipasvir/sofosbuvir for the treatment of HIV/HCV coinfected patients in October 2015, the first generation DAA regimens were left aside and became second line or alternative drugs in the treatment of hepatitis C. By evaluating the treatment patterns over 4 years in our clinical practice, we could clearly notice that the implementation of the guideline recommendations was carried out by our HIV/HCV providers in a proper manner. Our study also showed a trend for a decreasing degree of treatment failures in period 2 (0% vs 8.1%, p=0.006). The reason of this finding is probably the higher effectiveness of Ledipasvir/sofosbuvir; however, the design of our study does not allow an adequate evaluation of this parameter.”
2. Comment: The sensitivity of HCV RNA determination, which test systems were used, was not specified.
Answer: We have included the type and sensitivity of the test in the methods section. In the lines 82 to 84, the following sentences were added:
“To measure the HCV viral load, we used a real-time PCR (Roche HCV TaqMan® assay), which quantitates HCV RNA over a range of 15 to 100,000,000 UI/mL. Results below 15 UI/mL were reported as undetectable.”
3. Comment: p.4, line 116. Abbreviations AST, ALT are introduced, and there are no references in the text.
Answer: We have deleted those abbreviations.
Thanks a lot.
Sincerely,
Jose A. Gonzales Zamora
Corresponding author
Reviewer 2 Report
This is a short but interesting and meaningful manuscript. I do have a few questions.
Please consider moving the aims from the beginning of the “Methods” section to the end of the “Introduction”.
What is the rationale for dividing the study in two 2-year periods? Why not dividing in four 1-year periods?
For comparing (by chi-square) period 2 with the “rest of the study cohort”. Did you mean comparing period 1 with period 2?
Liver biopsies (done on ~40% of patients) were done with what purpose? Was this part of the diagnostic approach? If so, why only done in 40%.
Author Response
Dear Reviewer:
Thanks for your positive comments. I added the modifications you suggested. The changes are typed in red in the manuscript. The answers to your queries are the following:
1. Comment: Please consider moving the aims from the beginning of the “Methods” section to the end of the “Introduction”
Answer: We have moved the aims from “Methods” to “Introduction”
2. Comment: What is the rationale for dividing the study in two 2-year periods? Why not dividing in four 1-year periods?
Answer: The treatment guidelines for hepatitis C were updated at the end of 2015 and since the main objective of the study was to determine how these guidelines were implemented in our clinical practice, we decided to divide the study period in 2 groups: before the guideline update (2014-2015) and after (2016-2017). Another important factor to divide the study in this way was the approval of Ledipasvir/sofosbuvir for HIV/HCV coinfected individuals in October 2015, which expected to change the choice of antivirals among HIV providers from 2016 to 2017. We have mentioned these reasons in the first paragraph of our discussion (lines 124-141).
3. Comment: For comparing (by chi-square) period 2 with the “rest of the study cohort”. Did you mean comparing period 1 with period 2?
Answer: Yes, we phrased it in a different way to make it more understandable (lines 72-73)
4. Comment: Liver biopsies (done on ~40% of patients) were done with what purpose? Was this part of the diagnostic approach? If so, why only done in 40%
Answer: Liver biopsies are done to evaluate the degree of liver fibrosis prior to hepatitis C treatment, not for diagnostic purposes. In recent years, other methods have come out to assess liver fibrosis. One of these methods is transient elastography. Our study has shown that the use of transient elastography has increased in recent years given its non-invasive nature and lack of complications. We have included this explanation in the discussion (lines 168-173).
Thanks a lot
Sincerely,
Jose A. Gonzales Zamora
Corresponding author
Round 2
Reviewer 1 Report
Thank you, I am satisfied with the answers to my comments.